# Preparation and Properties of Double-Crosslinked Hydroxyapatite Composite Hydrogels

**DOI:** 10.3390/ijms23179962

**Published:** 2022-09-01

**Authors:** Benbo Zhao, Mingda Zhao, Liming Li, Shixiong Sun, Heping Yu, Yuan Cheng, Yuedi Yang, Yujiang Fan, Yong Sun

**Affiliations:** 1School of Chemical Engineering and Technology, North University of China, Taiyuan 030051, China; 2National Engineering Research Center for Biomaterials, Sichuan University, Chengdu 610064, China; 3Dezhou Graduate School, North University of China, Dezhou 253034, China; 4Guangdong Provincial Key Laboratory of Natural Rubber Processing, Agricultural Products Processing Research Institute, Chinese Academy of Tropical Sciences, Zhanjiang 524001, China

**Keywords:** hyaluronic acid, gelatin, dopamine, double crosslinked, hydroxyapatite-enhanced hydrogel

## Abstract

Natural polymer hydrogels have good mechanical properties and biocompatibility. This study designed hydroxyapatite-enhanced photo-oxidized double-crosslinked hydrogels. Hyaluronic acid (HA) and gelatin (Gel) were modified with methacrylate anhydride. The catechin group was further introduced into the HA chain inspired by the adhesion chemistry of marine mussels. Hence, the double-crosslinked hydrogel (HG) was formed by the photo-crosslinking of double bonds and the oxidative-crosslinking of catechins. Moreover, hydroxyapatite was introduced into HG to form hydroxyapatite-enhanced hydrogels (HGH). The results indicate that, with an increase in crosslinking network density, the stiffness of hydrogels became higher; these hydrogels have more of a compact pore structure, their anti-degradation property is improved, and swelling property is reduced. The introduction of hydroxyapatite greatly improved the mechanical properties of hydrogels, but there is no change in the stability and crosslinking network structure of hydrogels. These inorganic phase-enhanced hydrogels were expected to be applied to tissue engineering scaffolds.

## 1. Introduction

Hydrogels are a type of three-dimensional polymer network that have a large number of pore structures; hence, the exchange and circulation of substances becomes easy [1,2]. Natural polymer hydrogels with good biocompatibility and biodegradation can support cell adhesion, proliferation, and differentiation on its surface or inside. Thus, they are often used in the field of tissue engineering, for example, in cartilage repair and skin dressing [3,4,5,6]. However, their applications were hindered due to most natural polymers’ poor strength and toughness. Inorganic materials with good biocompatibility, such as hydroxyapatite and calcium phosphate, are widely used in bone tissue regeneration [7,8,9]. The introduction of inorganic materials into natural polymerbased hydrogels can effectively improve the mechanical properties of these hydrogels and provide a corresponding biological function of inorganic material [10,11].

Hyaluronic acid (HA), a natural linear polysaccharide, is a component of an extracellular matrix and plays an important role in biological processes (e.g., cell proliferation, cell differentiation, and morphogenesis) [12,13,14]. The HA molecules have many active groups, which can form a covalently cross-linked hydrogel network through chemical modification. Chen et al. introduced a sulfhydryl group into the HA molecular chain through an amidation reaction, which combined with collagen to form HA/Col composite hydrogels that could induce chondrogenic differentiation of cells [15]. Gelatin (Gel), as a derivative of collagen denaturation hydrolysis, shows excellent biocompatibility and biodegradability. Gel exists within a large number of bioactive sequences, such as arginine–glycine cell-binding sites, which is conducive to cell adhesion and spreading [16,17]. Additionally, Gel molecule chains with a large number of amino acid residues are easy to chemical modification. For example, methylacrylylated Gel (GelMA) could be obtained by the reaction between methacrylate anhydride and amino in Gel, which could form hydrogel through photoinitiated free radical polymerization, and have been widely used in tissue engineering scaffolds due to its simple and controllable manufacturing method [18,19,20].

Inspired by the adhesion chemistry of marine mussels, various hydrogels formed by derivatives with catechol groups have wide application prospects in tissue engineering scaffolds [21,22]. Studies indicate that catechol groups have a strong affinity for various organic and/or inorganic surfaces and biomacromolecules that can improve hydrogels’ mechanical properties [23,24,25]. As catechol and amino acids derivatives, dopamine (DA) has good adhesion and biocompatibility. Hence, they can be easily modified to natural polymers. Li et al. prepared DA-modified HA, which could fabricate a hybrid crosslinking hydrogel with type I collagen. This hybrid hydrogel can recruit autologous stem cells and induce differentiation of stem cells into chondrocytes [26].

In this study, we first simultaneously introduced the double bond and catechin into HA to form a photo-oxidized double-crosslinked hydrogel (HG) with GelMA. Then, hydroxyapatite was added to HG to form hydroxyapatite-enhanced hydrogels (HGH). Firstly, HA and Gel were modified with methacrylate anhydride to obtain hyaluronic acid methacrylate (HM) and GelMA with different substitution degrees, and HM was further modified with DA (HMD). The structure of HM, HMD, and GelMA was characterized by nuclear magnetic hydrogen spectroscopy (^1^H-NMR), Fourier-transform infrared spectroscopy (FT-IR) and Gel permeation chromatography (GPC). Photo-oxidized double-crosslinked hydrogels were prepared by photo-crosslinking of double bonds and oxidative-crosslinking of catechins. Moreover, hydroxyapatite, as an inorganic phase, was added to HG to improve their mechanical properties. To investigate the effects of the crosslinking degree and inorganic phase on hydrogels, the microstructure, mechanical properties, swelling, and degradation of hydrogels were tested. These inorganic phase-enhanced hydrogels provide a simple method for the design of tissue engineering scaffolds

## 2. Results and Discussion

### 2.1. Synthesis and Characterization of HMD and GelMA

Figure 1A presents the synthetic route of HMD and GelMA. First, HM with different substitutions was prepared by the reaction between methacrylate anhydride and hydroxyl of HA in different proportions. The carboxyl group in HM were activated by EDC/NHS and reacted with DA to obtain HMD. GelMA was prepared by an amidation reaction. Specifically, HM and HMD form a new vibration absorption peak at 1734 cm^−1^, which is the absorption vibration characteristic peak of -C=O- in the ester bond [27]. Additionally, the absorption peak of C-H on the benzene ring appeared near 3102 cm^−1^, and the characteristic peak of the amide bond appeared at 1640 cm^−1^. These findings revealed that the double bond and DA were successfully grafted into HA (Figure 1B). As shown in Figure 1C, the characteristic absorption peaks of the amide bond at 1642, and 1550 cm^−1^ were retained in GelMA. This indicated that the modification method was moderate and did not damage the integrity of the peptide chain. The changes in the absorption peak were not evident due to the overlap of the absorption peaks between the Gel and methyl propylene groups. Figure 2A presents the UV spectrum of HMD. Unlike HA and HM, HMD had a strong absorption peak at 280 nm which belongs to the catechin group. Simultaneously, no absorption peak was observed above 300 nm, indicating that DA was successfully grafted without oxidation.

The relative molecular weight and molecular weight distribution of HMD were determined through GPC (Figure 2B). The relative molecular weight of HA was greatly decreased after twice modification (223 kDa of HA and approximately 60 kDa of HMD) due to the hydrolysis of HA under the acidic reaction conditions when it was reacted with DA.

The product was characterized through ^1^H-NMR (Figure 2C,D), and the substitution degree of methacrylate anhydride and DA were calculated. Two new resonance peaks of HMD at 5.68 ppm and 6.13 ppm were the characteristic peaks of olefin proton in methacrylate anhydride. The substitution degree of methacrylate anhydride can be calculated by integrating the olefin proton peak with the characteristic peak of methyl H on N-acetylglucosamine at about 1.88 ppm (18%, 36%, and 52% for HMD-1, HMD-3, and HMD-5, respectively). Additionally, new peaks at 6.5–7.2 ppm were the characteristic peaks of the benzene ring in DA. The substitution degree can be calculated by integrating the characteristic peak of catechol at 6.5–7.2 ppm with the characteristic peak of methyl hydrogen on N-acetylglucosamine at about 1.88 ppm (the substitution degree for DA was approximately 10%). The DA substitution degree in the derivatives was also determined by UV spectroscopy using a calibration curve. All above results were list in the Table 1. Unlike Gel, GelMA showed two characteristic peaks at 5.31 ppm and 5.55 ppm, which were the chemical signals of hydrogen nucleus vibration that belongs to -C=CH2 in methacrylate anhydride. Additionally, the specific characteristic peak of methyl proton in methyl acrylamide was increased at 1.9 ppm [28]. Based on the change in the characteristic peak area at 2.89 ppm before and after the reaction, the degree of methacrylate anhydride was 54% by quantitative calculation. These indicated that HMD with different graft rates of methyl propylene groups and GelMA were successfully prepared.

### 2.2. Preparation and Microstructure of Hydrogels

Three HG hydrogels with different crosslinking density were prepared via UV photo-crosslinking by methyl propylene groups and oxidation-crosslinking by catechol. They are called HG-1, HG-3, and HG-5 according to the content of methacrylate anhydride in HMD. Similarly, hydroxyapatite was introduced to prepare three HGH hydrogels with different crosslinking density, called HGH-1, HGH-3, and HGH-5, respectively (Figure 3A).

After freeze-drying, the internal pore structures of the six groups of hydrogels were observed by SEM (Figure 3B). Image J was used to calculate the porosity of all hydrogels (Figure 3C,D). Each hydrogel formed interconnected porous structures with thick pore walls and a microfibril structure on the surface, which might be a three-dimensional structure formed by the complexation between the oxidative crosslinking of the DA and Gel molecules [29]. Meanwhile, the hydroxyapatite in HGH closely adhered to the pore wall, providing a basis for enhancing the mechanical properties of the hydrogel [30,31]. With an increase in the substitution degree of methacrylate anhydride in HMD, the porosity decreased gradually. Further, the crosslinking density of the hydrogel increased (HG-1 was approximately 50.6%, HG-3 was approximately 47.5%, and HG-5 was approximately 43.8%). The porosity of HGH-1, HGH-3, and HGH-5 is approximately 58.8%, 56.8%, and 58.2%, respectively. The interaction between hydroxyapatite and DA improved the pore structure of the hydrogel.

Additionally, the XRD result of HGH showed that the diffraction peaks of (100), (002), and (211) crystal planes appeared at 10.8°, 25.8°, and 31.7°, respectively. Meanwhile, the diffraction peaks of (112), (300), and (202) appeared near 32° at 39.8°, 46.7°, and 49.5°, respectively, and the diffraction peaks of (130), (222), and (213) crystal surfaces were observed (Figure 3E). The diffraction peaks of crystal planes were consistent with the standard data card of hydroxyapatite XRD diffraction (PDF#74-0565). This indicated that the crystal structure of hydroxyapatite does not change after the composite of hydroxyapatite into the hydrogel.

### 2.3. Mechanical Properties of Hydrogels

The compression properties of the two types of hydrogels were tested by using a universal tensile testing machine (Figure 4). In the compression process, with an increase in the methacrylate anhydride substitution in HMD, the crosslinking density of the hydrogel polymer network also increased, and additional force was required during compression due to the restricted movements of the polymer chain. The compression modulus of HG increased (HG-1: 9 kPa, HG-3: 14 kPa, and HG-5: 29 kPa). However, due to an increase in irreversible covalent crosslinking of the double bond, hydrogels were more fragile, and their compressive strength was decreased (HG-1: 138 kPa, HG-3: 85 kPa, and HG-5: 49 kPa) [32]. The compressive strength and compressive modulus of HGH were increased after hydroxyapatite was added (the compressive strength of HGH-1 increased to 254 kPa, and the compressive modulus of HGH-5 increased to 50 kPa). Hydroxyapatite could increase the density of hydrogel crosslinking network through interaction with hydrogel matrix, such as hydrogen bond interaction, thus improving the strength of hydrogel [33,34]. In addition, catechol groups have a strong affinity to hydroxyapatite [24,35]. Therefore, the mechanical properties of hydrogel were enhanced by the interaction between catechol and hydroxyapatite.

### 2.4. Swelling and Degradation of Hydrogels

Both HG and HGH have good hydrophilicity and reach a swelling equilibrium in approximately 10 h (Figure 5A,B). With the increased degree of methacrylate anhydride substitution, the hydrogel network becomes denser, resulting in a decrease in the equilibrium swelling rate (HG-1: 632%, HG-3: 502%, HG-5: 482%, Figure 5A). Further, the hydroxyapatite interacted with the polymer and increases the cross-linking density of the hydrogels, which endowed the hydrogel with higher porosity, thus improved the swelling performance of hydrogels to a certain extent (HGH-1: 754%, HGH-3: 647%, and HGH-5: 567%, Figure 5B). Additionally, the dense polymer network made the hydrogel structure more stable and the degradation rate (DR) lower. Particularly, HG-5 was completely degraded on the 13th day, but HGH-5 was not completely degraded on the 14th day (Figure 5C,D). The hydroxyapatite further improved the stability of the hydrogel.

## 3. Materials and Methods

### 3.1. Materials

HA (*M*_w_ = 340 kDa) was purchased from Shandong Furuda Biotechnology Company (Linyi, China). Gelatin (from cowhide) is purchased from Amresco (USA). Methacrylate anhydride (MA, 94%), 1-ethyl-3-(3-dimethylaminopropyl) carbodiimide (EDC, 98.5%), and N-hydroxysuccinimide (NHS, 99%) were purchased from Shanghai Maclean Biochemical Technology Co., Ltd. (Shanghai, China). DA, phenyl (2, 4, 6-trimethylbenzoyl) lithium phosphate (LAP, 98.0%) and sodium periodate were purchased from Sigma-Aldrich (St. Louis, MO, USA). N, N-dimethylformamide (DMF, 99.5%) and hyaluronidase (563 unit/mg) were obtained from Aladdin (Shanghai, China).

### 3.2. Synthesis of HA Derivatives (HM and HMD) and GelMA

HMD were synthesized in two steps per the method previously reported [36]. First, HM was prepared by modifying HA with methacrylate anhydride. Specifically, 600 mg HA was completely dissolved in ultra-pure water at a concentration of 2% *w*/*v*. Second, DMF was dropped into the HA solution (water: DMF = 3:2 *v*/*v*), and the mixture was cooled to 4 °C. Third, methacrylate anhydride was added to the mixture (The molar ratio between methacrylic anhydride and carboxylic groups of HA was 1, 3 and 5). Particularly, the pH value was maintained between 8 and 9 for 4 h by 0.5 M of NaOH solution, and the reaction was maintained at 4 °C for 20 h. The polymer was precipitated by adding ethanol into this solution and then dissolved in ultra-pure water. It was further purified via dialysis against ultra-pure water for 72 h (MWCO = 14 kDa). Finally, the solution was freeze-dried for 72 h to obtain HM, which was stored at −20 °C for later use.

HM was further modified by DA to prepare HMD. HM was completely dissolved in ultra-pure water at a concentration of 1% *w*/*v*. The pH value was adjusted to approximately 5.5 by 0.5 M HCl, and nitrogen was continuously introduced into the solution. EDC and NHS were added to the HM solution (molar ratio of HM: EDC: NHS = 1:3:3), and then the solution was stirred continuously for 30 min. DA hydrochloride (equal to the amount of the EDC substance) was added to the mixture. The reaction was stirred overnight at room temperature, maintaining a pH of 5.5 throughout the process. The solution was dialyzed in an acidic aqueous solution (HCl solution with pH 5.5) for 72 h (MWCO = 14 kDa). Finally, the solution was freeze-dried for 72 h to obtain HMD, which was stored at −20 °C for later use.

GelMA was synthesized as the method of previously reported [37]. At 50 °C, 1 g of type A Gel was dissolved into 10 mL phosphate buffer saline (PBS). Methacrylate anhydride was dropped into the Gel solution at a rate of 0.5 mL/min at 0.1 mL/g of Gel under continuous agitation. The mixture was allowed to react at 50 °C for 3 h. Finally, it was diluted five times with warm PBS at 40 °C to stop the reaction. The solution was dialyzed for 72 h in ultra-pure water (MWCO = 14 kDa) at 40 °C to remove unreacted methacrylate anhydride. Finally, it was freeze-dried for 72 h to obtain GelMA, which was stored at −20 °C for later use.

### 3.3. Characterization of HMD and GelMA

FT-IR: An appropriate amount of the product (HA, HM, HMD-1, HMD-3, HMD-5, Gel and GelMA) was mixed with potassium bromide and ground into fine powder. A Fourier infrared spectrometer (TENSOR27, BRUKER, Billerica, MA, USA) was used to scan the FT-IR spectrum at a range of 400–4000 cm^−1^.

UV–visible spectroscopy test (UV–vis): Catechol was determined via UV spectrophotometry. HA, HM, HMD-1, HMD-3, HMD-5 and DA were dissolved in 2 mg/mL ultra-pure water to prepare the sample solution, respectively. The UV–vis absorption spectra of the solution were measured in the range of 200–400 nm.

^1^H-NMR: The ^1^H-NMR spectrum was determined by dissolving HA, HM, HMD, Gel, and GelMA in D_2_O at a concentration of 2% (*w*/*v*) and by measuring using a magnetic resonance spectrometer (Bruker-600 NUCLEAR; Bruker, Fällanden, Switzerland). All ^1^H NMR spectra refer to the peak values of residual proton impurities in D_2_O at δ = 4.75 ppm.

GPC: It was used to test the molecular weight and distribution of the molecular weight of the material. The detector was the RID-20 differential refractive index from Shimadzu Company in Japan. The column was TSKgel GMPWXL aqueous gel from TOSOH Company in Japan. The mobile phase was the 0.1 N NaNO_3_ + 0.06%NaN_3_ aqueous solution at a flow rate of 0.6 mL/min, with a column temperature of 35 °C. Using polyethylene glycol as calibration, the sample was dissolved at a concentration of 3–5 mg/mL, with an injection volume of 20 μL.

### 3.4. Preparation of HA/Gel Hydrogel

Three groups of HMD samples were dissolved in ultra-pure water at a concentration of 2% *w*/*v*, respectively. Then, at 50 °C, GelMA was dissolved in the above solution to get the solution which contains 2% *w*/*v* of HMD and 5% *w*/*v* GelMA, and 0.5% photoinitiator phenyl (2, 4, 6-trimethylbenzoyl) lithium phosphate was added to the solution and mixed evenly. The double-crosslinked network hydrogels (HG) were obtained by pouring the precursor solution into the silica gel mold and exposing it to a 365 nm UV lamp for 5 min. Subsequently, a 5% sodium periodate solution was poured on the surface of the hydrogel to oxidize the catechin group. Additionally, HGH was obtained by adding 5% *w*/*v* hydroxyapatite before photo-crosslinking in the same manner.

### 3.5. Hydrogel Morphology and Structure

After freeze-drying, the hydrogel was immersed in liquid nitrogen before being truncated. The cross-section of the hydrogel was gilded by sputtering. Subsequently, its morphology and internal structure were observed by using a scanning electron microscope (SEM, HITACHI S-800, Tokyo, Japan). Image J software was used to calculate the porosity of the hydrogel based on the SEM image.

### 3.6. XRD

An X-ray diffractometer (BRUKER, Billerica, MA, USA) was used to analyze the XRD patterns of the hydrogel using CuKa characteristic radiation (wavelength λ = 0.154 nm, voltage 40 kV, 50 mA, scanning speed 1°/min, 2θ range 5–80°).

### 3.7. Mechanical Properties of Hydrogels

INSTRON 2345 compression test was conducted to evaluate the mechanical properties of hydrogels. The hydrogel with a diameter of 10 mm and a thickness of 3 mm was tested to determine its compression performance. The sample was placed in a testing machine and compressed at a constant speed of 1 mm/min until the sample broke. The slope of the first 10% linear part of the obtained stress–strain curve was regarded as the compression modulus. Three samples were tested in each group.

### 3.8. Swelling Property of Hydrogels

After freeze-drying, the original weight W_0_ of each hydrogel was recorded. Subsequently, the hydrogel was placed in PBS solution at 37 °C to observe its swelling changes. The samples were obtained at 2, 6, 12, 24, 48, and 72 h using filter paper to absorb water on the surface of the hydrogel, and the weight W_i_ of the hydrogel sample was recorded. Three samples were tested in each group. If the mass does not change, the swelling equilibrium of the hydrogel is achieved. The following formula is used to calculate the swelling degree (W) of the hydrogel:Swelling degree (W) = (W_i_ − W_0_)/W_0_ × 100%.

### 3.9. Degradation In Vitro

After freeze-drying each group of hydrogels, the original weight M_0_ was recorded. Subsequently, the hydrogels were placed in a PBS degradation solution containing 100 units/mL of hyaluronidase and incubated at 37 °C. Three samples were tested in each group. The degradation solution was replaced daily, and the hydrogel was removed at a predetermined time, freeze-dried, and weighed again, and its mass M_i_ was recorded. The hydrogel’s degradation rate (DR) can be calculated by using the following formula:Degradation rate (DR) = (M_0_ − M_i_)/M_0_ × 100%.

## 4. Conclusions

In this study, HG based on HA and Gel derivatives was prepared through photo-crosslinking of double bonds and oxidative-crosslinking of catechins. Meanwhile, HGH was also prepared by introducing hydroxyapatite to HG. With an increase in the substitution degree of the methyl propylene group in HMD, the HG crosslinking density increases. This leads to lower porosity, mechanical strength increasing, brittleness increasing, equilibrium swelling rate decreasing, and degradation time becoming longer. Hydroxyapatite retained the pore structure of the hydrogel and greatly improved the mechanical properties and stability of hydrogel scaffolds to a certain extent. In general, we prepared hydrogels with controllable properties by changing the substitution degree of precursors, and thus improved their properties by introducing inorganic phases. This simple preparation strategy for hydrogels is expected to be applied to tissue engineering scaffolds.

## Figures and Tables

**Figure 1 ijms-23-09962-f001:**
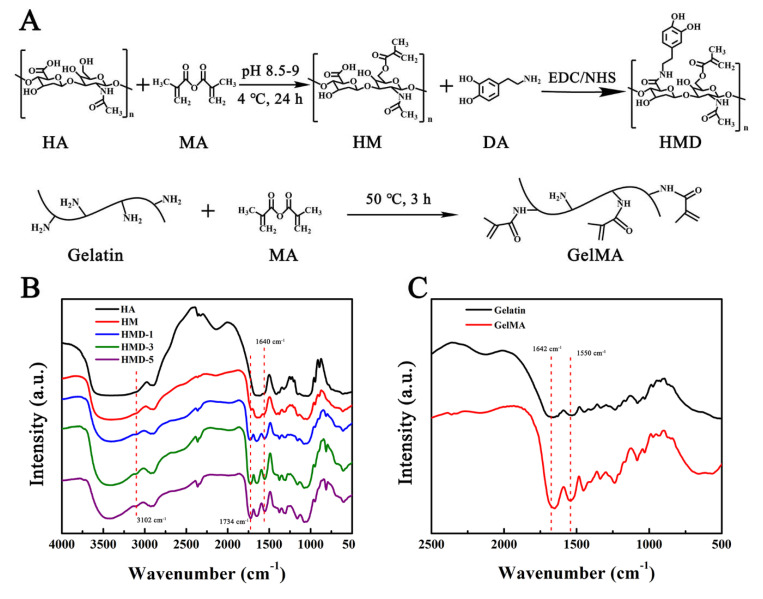
(**A**) Synthetic route of HGD and GelMA. (**B**) FT-IR of HA, HM, HMD-1, HMD-3 and HMD-5. (**C**) FT-IR of gelatin and GelMA.

**Figure 2 ijms-23-09962-f002:**
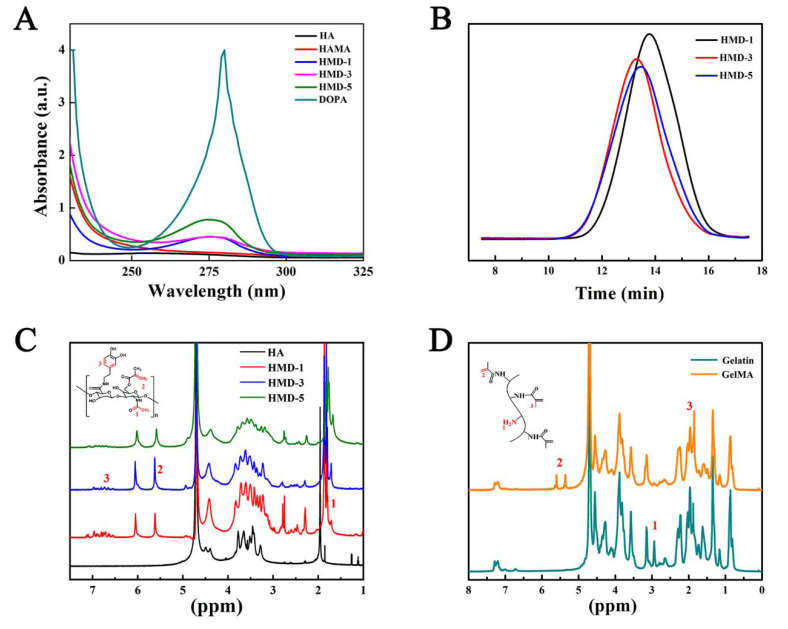
(**A**) UV–vis of HMD. (**B**) Molecular weight distribution of HMD. (**C**) ^1^H NMR of HMD. (**D**) ^1^H NMR of GelMA.

**Figure 3 ijms-23-09962-f003:**
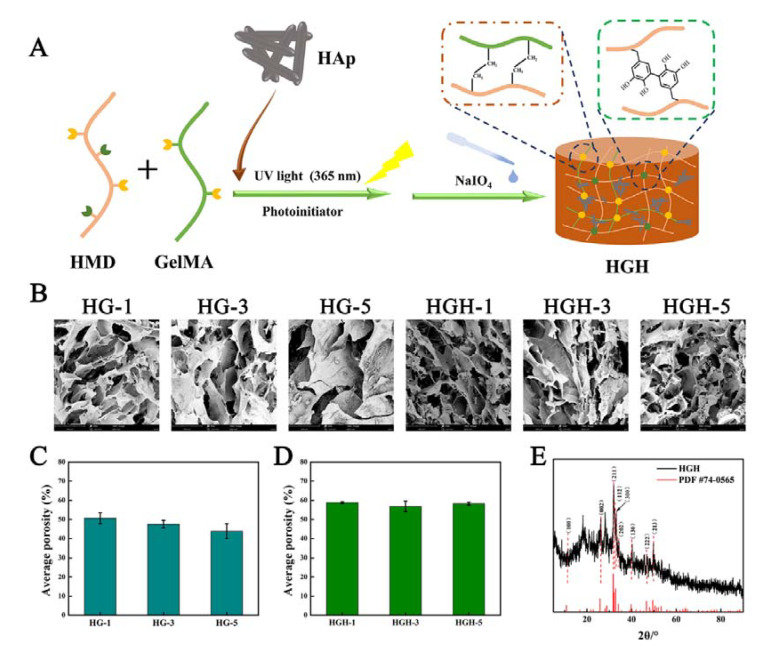
(**A**) Schematic diagram of the hydrogel. (**B**) SEM of HG and HGH hydrogels. (**C**,**D**) Average porosity of HG and HGH hydrogels calculated by using Image J. (**E**) The XRD result of HGH.

**Figure 4 ijms-23-09962-f004:**
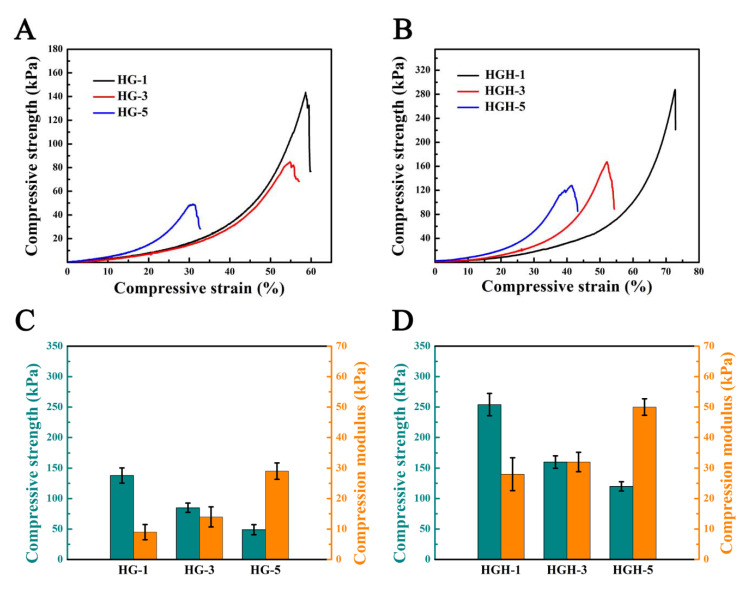
(**A**,**B**) Compressive stress–strain curves of HG and HGH hydrogels. (**C**,**D**) Histogram of the compressive strength and compressive modulus of HG and HGH hydrogels.

**Figure 5 ijms-23-09962-f005:**
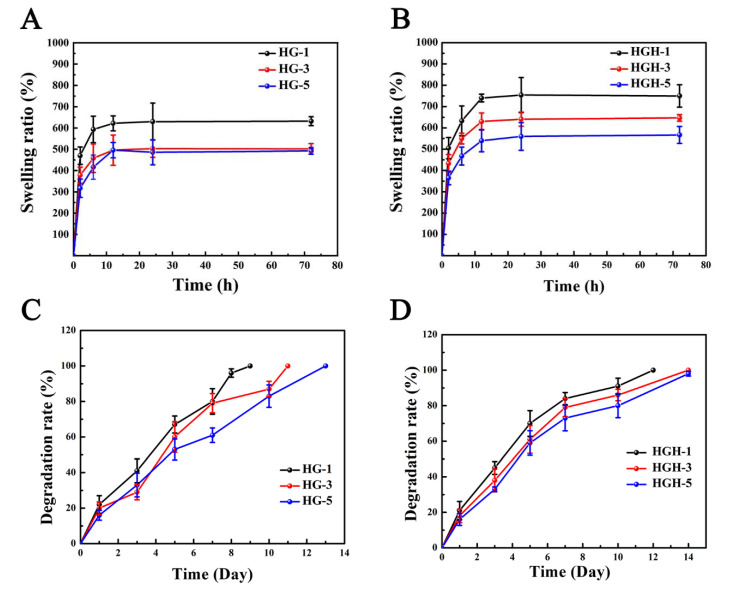
(**A**,**B**) Swelling curve of HG and HGH hydrogels. (**C**,**D**) Degradation curve of HG and HGH hydrogels.

**Table 1 ijms-23-09962-t001:** The substitution degree of methacrylate anhydride and DA in HMD-1, HMD-3 and HMD-5.

	Methacrylate Anhydride (By ^1^H-NMR)	DA (By ^1^H-NMR)	DA (By UV Spectroscopy)
HMD-1	18%	9.3%	6.3%
HMD-3	36%	9.6%	6.1%
HMD-5	52%	10.1%	9.4%

## Data Availability

Data is contained within the article.

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
