# Peer review of "Preparation and Properties of Double-Crosslinked Hydroxyapatite Composite Hydrogels"

_ijms, 2022, doi:10.3390/ijms23179962_

Round 1

Reviewer 1 Report

The paper present the synthesis of hydrogels based on hyaluronic acid and gelatin and their characterization from the point of view of the mechanical properties, swelling and degradation in the presence of hyaluronidase. The hydrogels were obtained based on the crosslinking of the double bounds attached to the both polymers and the oxidative crosslinking of the dopamine groups attached to the hyaluronic acid. Some aspects must be improved.

The role of the oxidative crosslinking must be demonstrated by comparison the properties of the hydrogels obtained by double cross-linking (HG1, HG3 or HG5 sample) with the hydrogels obtained only by photo-polymerization obtained using HM derivative (hyaluronic acid modified with methacrylic anhydride).

The novelty of the present work must be evidenced.

A table can be introduced to present the substitution degree of HMD-1, HMD3 and HMD-5 derivatives with methacrylic anhydride and with dopamine. From the NMR spectra it seems that HMD-1 have a higher content of DA compared to HMD-5. Dopamine content in the derivatives can be also determined by UV spectroscopy using a calibration curve and the results can be introduced in this table.

A possible explanation for the improved swelling performances of the hydrogels containing hydroxyapatite) compared with the hydrogels without inorganic filler must be introduced (line 187).

Some small aspects must be corrected and some data must be introduced:

Figure 1A: The abbreviation DA must be used instead of DOPA.

2C: The notation of the protons 1, 2 and 3 cannot be seen in the chemical structure.

Figure 3B: the notation of the samples must be modified (HG3 and HG5 instead of HG2 and HG3).

Line 189: What represent the abbreviation “DR”?

Line 212: The molar ratio between methacrylic anhydride and carboxylic groups of HA was 1, 3 and 5 in the synthesis?

In the Chapter 3.4, Preparation of HA/Gel hydrogels, the gravimetric ratio between the polymers or the volumetric ratio between the two solution should be introduced. How the penetration of the sodium periodate solution inside the hydrogel was ensured in order to obtain an oxidative crosslinking of all the DA groups?

For the mechanical properties and SEM analysis, the hydrogels were washed with water after the obtaining or not? All the hydrogels were first freeze-dried including the hydrogels used in the compression tests?

Some sentences must be modified:

”the stiffness of hydrogels with becomes higher” (line 21);

“Additionally, Gel molecule chains with a large volume of amino acid residues that are easy to chemical modification; methylacrylylated Gel (GelMA) can form hydrogel through photoinitiated free radical polymerization…” (lines 53-56)

“Dopamine (DA) have good adhesion and biocompatibility as they are catechol and amino acids derivatives. ”

“methacrylate anhydride” was used instead of “methacrylic anhydride”.

Author Response

Point 1: The role of the oxidative crosslinking must be demonstrated by comparison the properties of the hydrogels obtained by double cross-linking (HG1, HG3 or HG5 sample) with the hydrogels obtained only by photo-polymerization obtained using HM derivative (hyaluronic acid modified with methacrylic anhydride).

Response 1: From our previously works [1], the secondary covalent network formed by the oxidative crosslinking could coordinate with the primary photo-crosslinked network to improve the toughness of hydrogels and keep a higher strength.

[1] Zhao B. B.; Zhao M. D.; Sun H. G.; Yang Y. D.; Sun S. X.; Yu H. P.; He M. M.; Sun Y.; Cheng Y., Preparation and characterization of photo-oxidative dual-crosslinked chitosan/hyaluronic acid hydrogels, Reactive and Functional Polymers (2022), https://doi.org/10.1016/j.reactfunctpolym.2022.105378.

Point 2: The novelty of the present work must be evidenced.

Response 2: Methacrylate anhydride and catechin group were simultaneously introduced into the HA chain and formed photo-oxidized double-crosslinked hydrogels (HG) with GelMA. Then the hydroxyapatite was introduced into HG to form hydroxyapatite-enhanced hydrogels (HGH). The effects of different substitution degrees of methacrylate anhydride in HMD and the introduction of hydroxyapatite on the properties of hydrogels were detected. By controlling the substitution degree of methacrylate anhydride, the properties of hydrogels can be regulated; Besides, the catechin group could not only improve the toughness of hydrogel, but could also interact with hydroxyapatite, enhance the integration between polymer chain and inorganic particles.

[2] Lu, G. G.; Xu, Y.; Liu, Q. Y.; Chen, M. Y.; Sun, H.; Wang, P. L.; Li, X.; Wang, Y. X.; Li, X.; Hui, X. H.; Luo, E.; Liu, J.; Jiang, Q.; Liang, J.; Fan, Y. J.; Sun, Y.; Zhang, X. D., An instantly fixable and self-adaptive scaffold for skull regeneration by autologous stem cell recruitment and angiogenesis. Nat. Commun. 2022, 13, (1), 2499.

Point 3: A table can be introduced to present the substitution degree of HMD-1, HMD3 and HMD-5 derivatives with methacrylic anhydride and with dopamine. From the NMR spectra it seems that HMD-1 have a higher content of DA compared to HMD-5. Dopamine content in the derivatives can be also determined by UV spectroscopy using a calibration curve and the results can be introduced in this table.

Response 3: According to the suggestion, Table 1 was added in the revised manuscript including the substitution degree of methacrylic anhydride calculated by 1H-NMR and the substitution degree of dopamine calculated by 1H-NMR and UV spectroscopy in HMD-1, HMD3 and HMD-5. From the NMR spectra, the intensity of the peak did not represent the substitution degree, which was calculated by the ratio of the characteristic peak of catechol at 6.5–7.2 ppm and the characteristic peak of methyl hydrogen on N-acetylglucosamine at 2.03 ppm.

Point 4: A possible explanation for the improved swelling performances of the hydrogels containing hydroxyapatite) compared with the hydrogels without inorganic filler must be introduced (line 187).

Response 4: The hydroxyapatite interacted with polymers and increased the cross-linking density of the hydrogels, which endowed the hydrogels with higher porosity, thus improved the swelling performance of hydrogels to a certain extent. It had been added in the revised manuscript.

Some small aspects must be corrected and some data must be introduced:

Point 5: Figure 1A: The abbreviation DA must be used instead of DOPA.

Response 5: According to the suggestion, the abbreviation DA has been used in Figure 1A.

Point 6: 2C: The notation of the protons 1, 2 and 3 cannot be seen in the chemical structure.

Response 6: According to the suggestion, the notation of the protons 1, 2 and 3 was added in Figure 2C.

Point 7: Figure 3B: the notation of the samples must be modified (HG3 and HG5 instead of HG2 and HG3).

Response 7: According to the suggestion, the notation of the samples have been modified in Figure 3B.

Point 8: Line 189: What represent the abbreviation “DR”?

Response 8: The abbreviation “DR” represent degradation rate. According to the suggestion, we defined the acronyms of DR in the revised manuscript.

Point 9: Line 212: The molar ratio between methacrylic anhydride and carboxylic groups of HA was 1, 3 and 5 in the synthesis?

Response 9: Yes, in the synthesis, the molar ratio between methacrylic anhydride and carboxylic groups of HA was 1, 3 and 5. The expression in the manuscript has been revised.

Point 10: In the Chapter 3.4, Preparation of HA/Gel hydrogels, the gravimetric ratio between the polymers or the volumetric ratio between the two solution should be introduced. How the penetration of the sodium periodate solution inside the hydrogel was ensured in order to obtain an oxidative crosslinking of all the DA groups?

Response 10: HMD was firstly dissolved in ultra-pure water at a concentration of 2% w/v, then at 50 °C, GelMA was dissolved in the above solution to get the HMD/GelMA solution which contain 2% of HMD and 5% GelMA. We have modified the relevant statements in the revised manuscript. We controlled the action time of sodium periodate solution to be 1 min to ensure consistency for all samples.

Point 11: For the mechanical properties and SEM analysis, the hydrogels were washed with water after the obtaining or not? All the hydrogels were first freeze-dried including the hydrogels used in the compression tests?

Response 11: All hydrogels were washed with ultra-pure water after crosslinking. The hydrogels used for the compression tests were not freeze-dried.

Point 12: Some sentences must be modified:

”the stiffness of hydrogels with becomes higher” (line 21);

“Additionally, Gel molecule chains with a large volume of amino acid residues that are easy to chemical modification; methylacrylylated Gel (GelMA) can form hydrogel through photoinitiated free radical polymerization…” (lines 53-56)

“Dopamine (DA) have good adhesion and biocompatibility as they are catechol and amino acids derivatives. ”

“methacrylate anhydride” was used instead of “methacrylic anhydride”.

Response 12: According to the suggestion, the above sentences were modified in the revised manuscript.

"The revised manuscript" please see the attachment

Reviewer 2 Report

Review on manuscript ijms-1878849

The article is about the preparation and detailed characterization of double-crosslinked HAP composite hydrogel. 

The article is well-written and well-organized, but a few corrections are necessary before publication. 

My comments/suggestions: 

- Fig. 1: the figure caption is incomplete. Please give more information on samples. As Fig.1B please sign the main vibration frequnecies. 

- Fig. 5. for better comparison of the data, please use similar y-axes. e.g. swelling ratio from 0-1000 %

- Line 215. what was the pore size (cut-off) for dialysis?  

- for determination of average Mw did you try another techniques as well, e.g. rheology or DLS? 

Author Response

Point 1: Fig. 1: the figure caption is incomplete. Please give more information on samples. As Fig.1B please sign the main vibration frequnecies. 

Response 1: According to the suggestion, we completed the figure caption and signed the main vibration frequencies in Figure 1.

Point 2: Fig. 5. for better comparison of the data, please use similar y-axes. e.g. swelling ratio from 0-1000 %

Response 2: According to the suggestion, the y-axes in Fig. 5A was changed from “0-800 %” to “0-1000 %”.

Point 3: Line 215. what was the pore size (cut-off) for dialysis?  

Response 3: The molecular cut off of dialysis tube is 14 kDa. According to the suggestion, we added the explanation in the revised manuscript.

Point 4: for determination of average Mw did you try another techniques as well, e.g. rheology or DLS? 

Response 4: We used GPC for the determination of average Mw. From our previous study [1, 2], GPC is a powerful tool to determine the average Mw of polymers.

[1] Xu Y.; Lu G. G.; Chen M. Y.; Wang P. L.; Li Z. L.; Han X. W.; Liang J.; Sun Y.; Fan Y. J.; Zhang X. D., Redox and pH dual-responsive injectable hyaluronan hydrogels with shape-recovery and self-healing properties for protein and cell delivery, Carbohydr. Polym. 250 (2020) 116979.

[2] Ma M. C.; Chen Y. F.; Zhao M. D.; Sui J. H.; Guo Z. H.; Xu Z. Y.; Sun Y.; Fan Y. J.; Zhang X. D., Hierarchical responsive micelle facilitates intratumoral penetration by acid-activated positive charge surface and size contraction, Biomaterials 271 (2021) 120741.

"The revised manscript" please see the attachment.

Round 2

Reviewer 1 Report

The paper was improved.

Some English changes are still required, like: „methacrylate anhydride” or „was become higher”.